# Cat Scratch Disease—A Benign Disease with Thymic Hyperplasia Mimicking Lymphoma

**DOI:** 10.3390/diagnostics13142457

**Published:** 2023-07-24

**Authors:** Ming Hui Leong, Mohd Jadi Nabillah, Iqbal Hussain Rizuana, Abdullah Asma, Thean Yean Kew, Geok Chin Tan

**Affiliations:** 1Department of Radiology, Faculty of Medicine, Universiti Kebangsaan Malaysia, Kuala Lumpur 56000, Malaysia; 2Hospital Canselor Tuanku Muhriz, Jalan Yaakob Latif, Bandar Tun Razak, Kuala Lumpur 56000, Malaysia; 3Department of Otorhinolaryngology-Head & Neck Surgery, Faculty of Medicine, Universiti Kebangsaan Malaysia, Kuala Lumpur 56000, Malaysia; 4Department of Pathology, Faculty of Medicine, Universiti Kebangsaan Malaysia, Kuala Lumpur 56000, Malaysia

**Keywords:** cat scratch disease, lymphoma, cervical nodes, thymic hyperplasia, CT scan

## Abstract

Cat scratch disease (CSD) is a benign condition caused by the inoculation of Bartonella henselae. The imaging findings are non-specific, and it is difficult to diagnose the disease via imaging. However, imaging studies help exclude other differential diagnoses in diagnostic dilemmas. We encountered a case of a 17-year-old adolescent who presented with painful neck swelling. CT showed multiple bilateral cervical lymphadenopathies with triangular soft tissue mass at the anterior mediastinum likely to be thymic hyperplasia, which is unusual in CSD and was mistaken for a lymphoproliferative disorder. Tissue diagnosis with a thorough clinical history yielded the diagnosis of cat scratch disease, and follow-up imaging showed resolution of the cervical lymphadenopathy and thymic hyperplasia.

A 17-year-old adolescent presented with painful right-sided neck swelling for two weeks; this gradually increased in size and was associated with right otalgia. The patient had no history of ear discharge or skin redness and denied any history of fever or contact with tuberculosis patients. However, he had an episode of upper respiratory tract infection before he developed otalgia. Examination of the patient revealed multiple tender, enlarged bilateral cervical lymph nodes measuring up to 1 cm in size. Otoscopy examination revealed an intact tympanic membrane with no evidence of inflammation. No enlarged lymph nodes were palpable elsewhere, and there was no hepatosplenomegaly. Patient then proceeded with contrast-enhanced computed tomography (CECT) neck (Figure 1 and Figure 2).

**Figure 1 diagnostics-13-02457-f001:**
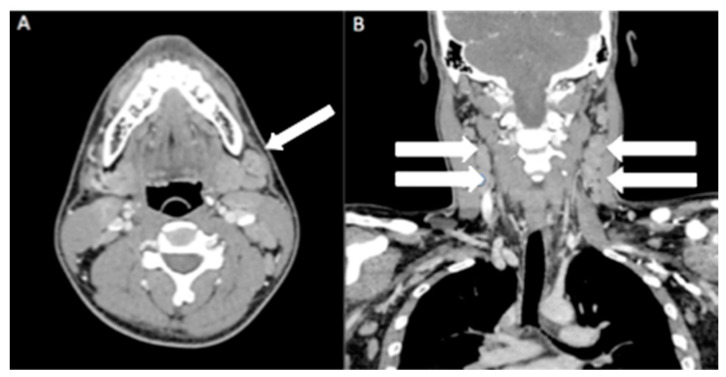
Contrast-enhanced computed tomography (CECT) neck on axial (**A**) and reconstructed coronal view (**B**) showed multiple bilateral enlarged cervical nodes (white arrows), with some nodes showing rounded configuration. There was no central necrosis or matted nodes.

**Figure 2 diagnostics-13-02457-f002:**
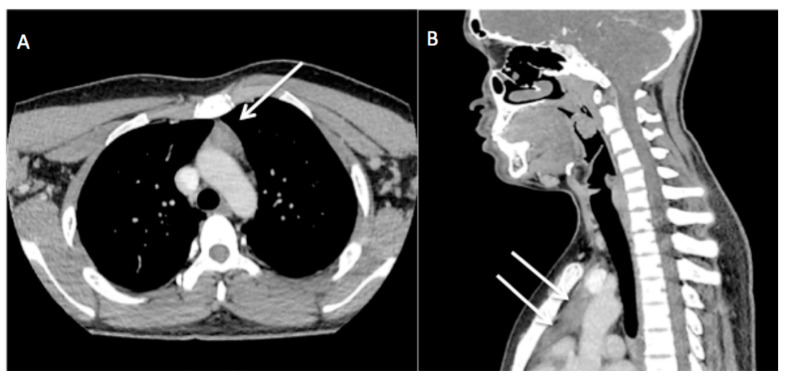
CECT neck on axial (**A**) and reconstructed sagittal view (**B**) showed triangular-shaped homogenous soft tissue lesion in anterior mediastinum (white arrow) with slight convexity to the left but no mass effect on the surrounding structure. A clinical suspicion of lymphoma with a differential diagnosis of tuberculosis was made. The anterior mediastinal mass was initially thought to be thymic hyperplasia, which could represent the lymphomatous involvement of the thymus. Subsequently, excision biopsy of the enlarged nodes was performed and sent for histopathological examination (HPE).

**Figure 3 diagnostics-13-02457-f003:**
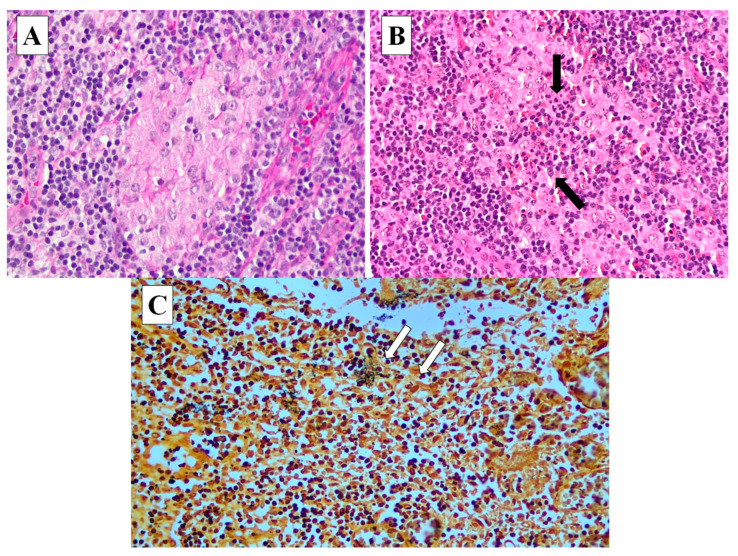
This section of the right neck nodes shows a mildly effaced nodal architecture with many small granulomas (**A**) throughout the nodal parenchyma. There are occasional granulomas demonstrating central microabscesses (black arrows) (**B**). Warthin–Starry silver staining shows clusters of bacterial microorganisms (white arrows) (**C**). Histologically, the right neck lymph node showed granulomatous inflammation with occasional microabscesses (Figure 3). In addition, a Warthin–Starry silver stain showed a few clusters of bacteria. Ziehl–Neelsen staining for acid-fast bacilli was negative. CD3 and CD20 immunohistochemistry showed a mixed population of B and T lymphocytes. The overall histological features coupled with the proper clinical history are consistent with cat scratch disease.

**Figure 4 diagnostics-13-02457-f004:**
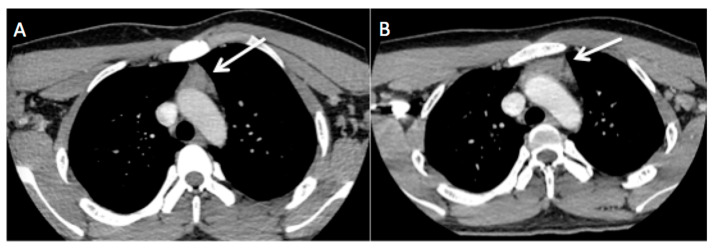
Initial CECT (**A**) and follow-up CECT 8 months later (**B**). Post-treatment, the size of the triangular-shaped soft tissue lesion (white arrow) reduced, representing thymic tissue. The patient completed 1 week of intravenous Augmentin, and his symptoms resolved completely. Imaging follow-up 8 months later showed an overall reduction in the size of the enlarged cervical nodes and thymic tissue (Figure 4 and Figure 5).

**Figure 5 diagnostics-13-02457-f005:**
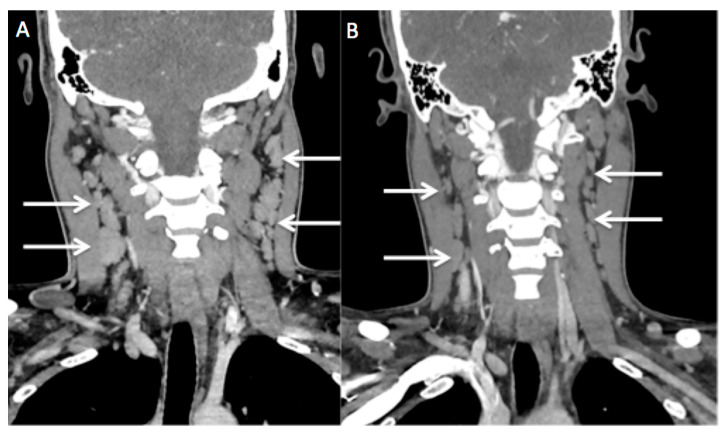
Initial CECT (**A**) and follow-up CT 8 months later (**B**). Post-treatment, the size of the bilateral enlarged cervical nodes reduced (white arrows). Classically, the risk factors described for CSD are being licked on the face, scratched, or bitten by a kitten [1]. The most common clinical presentation is regional lymphadenopathy, which becomes visible after one to three weeks of inoculation. Upper extremities and the head and neck have been described as the most common sites for regional lymphadenopathy in CSD. CSD is usually a self-limiting disease. Antibiotics are administered for complicated, severe disease or those with systemic manifestation [2,3]. Radiologic diagnosis of this disease is difficult due to its non-specific imaging findings. According to the literature, ultrasound of the regional lymphadenopathy may show enlarged lymph nodes with hyperemia being seen occasionally, with or without central necrosis. Suppurative lymphadenitis may develop in 10% of clinically identifiable lymphadenopathy [4]. History of cat exposure was not recalled by the patient initially. CECT scan in our patient revealed multiple bilateral enlarged cervical lymph nodes and anterior mediastinal mass, which is unusual in CSD; hence, lymphoma was concluded as the provisional diagnosis. However, excisional biopsy can exclude lymphoma, as was the case with our patient. Differential diagnoses of non-specific regional lymphadenopathy are broad. Therefore, a thorough clinical history with a high index of suspicion is necessary to achieve a correct diagnosis. Painless regional lymphadenopathy in the head and neck is one of the most common initial presentations of Hodgkin lymphoma. In lymphoma, the rounded appearance of the enlarged nodes is more common, although elliptical shapes can be seen [5]. Tuberculous lymphadenitis and Kikuchi disease are other differential diagnoses that need to be considered. As tuberculous lymphadenitis is a great mimicker of many diseases in the case of cervical lymphadenopathy, it would be one of the differential diagnoses, especially in Malaysia, where TB is endemic [6]. Kikuchi disease commonly presents with multiple enlarged lymph nodes, particularly in the head and neck, as well as the axilla, inguinal, and supraclavicular regions. It is associated with fever and muscle aches, as reported by Hayati et al. and Halimudin et al., which are similar symptoms to the ones our patient presented [7,8]. Tuberculous lymphadenitis is usually painless, whereas Kikuchi disease is painful lymphadenopathy accompanied by systemic symptoms [9]. The most common CECT findings of Kikuchi disease include lymphadenopathy with larger numbers of necrotic foci within the nodes; however, the number of foci is less extensive than the quantity observed in tuberculous lymphadenitis. The enlarged lymph nodes in Kikuchi disease also show significantly higher necrosis attenuation than tuberculous lymphadenitis. Calcification is less commonly observed [9]. The typical CECT findings of tuberculous lymphadenitis include central low-attenuation necrosis with peripheral rim enhancement with or without calcification [10]. However, these diseases are difficult to differentiate in daily practice due to overlapping findings on CECT. CSD typically presents as fever and regional lymphadenitis of the region of inoculation. The usual clinical manifestation is painful but self-limited lymphadenopathies, especially of the neck, axillary region, or upper extremities. Rarely, CSD presents as prolonged fever of unknown origin, without any lymphadenopathies (5–25%). More rarely (5–14%), there may be hematogenous spread causing systemic disease with atypical manifestations such as thymic hyperplasia, hepatic and splenic abscesses, endocarditis, and encephalopathy osteomyelitis [11]. Ultrasound is another helpful imaging modality used to assess cervical lymphadenopathy in doubtful cases and may provide useful information, aiding in diagnosing the disease. Imaging features that need to be assessed carefully include shape and homogeneity of the nodes, intranodal cystic necrosis, posterior enhancement, matting, and surrounding soft tissue edema [12]. We present this unique CSD case report describing a teenager with enlarged and painful neck nodes as well as an enlarged thymus. CSD is a common cause of lymphadenopathy, particularly in the young adult and pediatric population. Many disease states overlap in radiological manifestations when compared with CSD. Radiologists who are familiar with the imaging findings may suggest the diagnosis. A thorough clinical history and physical examination, as well as a high index of suspicion, will aid in the diagnosis of this disease.

## Data Availability

Not applicable.

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
