# Peer review of "Cat Scratch Disease—A Benign Disease with Thymic Hyperplasia Mimicking Lymphoma"

_diagnostics, 2023, doi:10.3390/diagnostics13142457_

Round 1
Reviewer 1 Report
An interesting case of CSD.
I would add histological images.
Author Response
Dear Reviewer 1,
We thank the reviewer very much for reviewing the manuscript and for your positive response.
Thank you for your suggestion. We have added the histological images as suggested.
Kindly refer Figure 3 and Line 52-68.
Best regards,
Authors
Reviewer 2 Report
The authors report a patient with cat scratch disease with thymic hyperplasia, which is potentially interesting for readers.
Major.
1. The reviewer thinks the diagnosis of CSD is not convincing in the current description. The authors should describe more the histological features, for example, the presence/absence of stellate abscess, a histological feature of CSD. In addition, the authors can try to detect the pathogen by Warthin-Starry staining or culture test.
2. Please discuss more the relationship between CSD and thymic hyperplasia.
Minor.
1. Page 3, lines 59-64: The font size is smaller than other lines.
2. Figure 1B. What do the two blue dots on one arrow indicate?
The English can be improved by native English speakers.
Author Response
Dear Reviewer 2,
We thank the reviewer very much for reviewing the manuscript and for your positive comments and suggestions. We have made every attempt to address the comments. We have taken every effort to improve on our English.
Suggestions:
1. The reviewer thinks the diagnosis of CSD is not convincing in the current description. The authors should describe more the histological features, for example, the presence/absence of stellate abscess, a histological feature of CSD. In addition, the authors can try to detect the pathogen by Warthin-Starry staining or culture test.
Thank you for your suggestion. We have added the histological images as suggested. We have also added in Warthin-Starry staining.
Kindly refer Figure 3 and Line 52-68
2. Please discuss more the relationship between CSD and thymic hyperplasia.
Thank you. We made the changes as suggested at Line 158-164
3. Page 3, lines 59-64: The font size is smaller than other lines.
Thank you. We have standardized the font sizes
4. Figure 1B. What do the two blue dots on one arrow indicate?
Thank you. We have removed the dots.
5. The English can be improved by native English speakers.
We have submitted to MDPI English Editing services and made the corrections accordingly.
Best regards,
Authors
Reviewer 3 Report
First of all, I would like to thank you for inviting me to review the manuscript entitled: Cat Scratch Disease- A Benign Disease With Thymic Hyperplasia Mimicking As Lymphoma
The presented case involves an important area of health and presents a clear and clinically useful message. The manuscript is well written in terms of clarity, style, and use of English and has a logical construction. The figures are of good quality and relevant to the clinical message. The references are appropriate and current.
Major issue:
1) The authors present a case in which the correct diagnosis was made by pathological examination. The inclusion of histopathological photos is necessary.
Minor issues:
1) In the title, the authors should erase “as” or use another verb such as “presenting as”.
2) On page 2, line 43, the authors mention that “Diagnosis of lymphoma with differential diagnosis of tuberculosis was made.” This should be changed to “Clinical suspicion” or “The differential diagnosis included” since the diagnosis was made by pathologic examination.
English language needs to be improved.
Author Response
Dear Reviewer 1,
The presented case involves an important area of health and presents a clear and clinically useful message. The manuscript is well written in terms of clarity, style, and use of English and has a logical construction. The figures are of good quality and relevant to the clinical message. The references are appropriate and current.
We thank the reviewer very much for reviewing the manuscript and for your positive comments and suggestions. We have made every attempt to address the comments. We have taken every effort to improve on our English.
Suggestions by Reviewer:
1. The authors present a case in which the correct diagnosis was made by pathological examination. The inclusion of histopathological photos is necessary.
Thank you for your suggestion. We have added the histological images as suggested. We have also added in Warthin-Starry staining.
Kindly refer Figure 3 and Line 52-68
2. In the title, the authors should erase “as” or use another verb such as “presenting as”.
Thank you. We have removed “as” from the title.
3. On page 2, line 43, the authors mention that “Diagnosis of lymphoma with differential diagnosis of tuberculosis was made.” This should be changed to “Clinical suspicion” or “The differential diagnosis included” since the diagnosis was made by pathologic examination.
Thank you. We have changed to "Clinical suspicion"
4. English language needs to be improved.
Thank you. We have submitted for MDPI English editing and made the recommended changes.
Best regards,
Authors
Round 2
Reviewer 2 Report
Thank you for addressing the reviewer's comments. Here, the reviewer would like to point out only small grammatical errors.
Figure 3: Main text Section of the right neck nodes; the reviewer does not understand the meaning of "Main text Section".
Line 123: thymic hyperplasia; perhaps thymic hyperplasia is correct.
Acceptable.
Author Response
Dear Reviewer (Round 2),
Thank you for addressing the reviewer's comments. Here, the reviewer would like to point out only small grammatical errors.
We thank the reviewer very much for reviewing the manuscript and for your positive response.
Thank you for your suggestion.
- Figure 3: Main text Section of the right neck nodes; the reviewer does not understand the meaning of "Main text Section".
We have rephrased it as "Figure 3: This section of the right neck nodes ..." - Line 123: thymic hyperplasis; perhaps thymic hyperplasia is correct.
Yes, thank you. It was a typo. Its now at Line 119 as "thymic hyperplasia, ...."
Best regards,
Authors